# Exploring Silk Sericin for Diabetic Wounds: An In Situ-Forming Hydrogel to Protect against Oxidative Stress and Improve Tissue Healing and Regeneration

**DOI:** 10.3390/biom12060801

**Published:** 2022-06-08

**Authors:** Sara Baptista-Silva, Beatriz G. Bernardes, Sandra Borges, Ilda Rodrigues, Rui Fernandes, Susana Gomes-Guerreiro, Marta Teixeira Pinto, Manuela Pintado, Raquel Soares, Raquel Costa, Ana Leite Oliveira

**Affiliations:** 1CBQF—Centro de Biotecnologia e Química Fina, Laboratório Associado, Escola Superior de Biotecnologia, Universidade Católica Portuguesa, 4169-005 Porto, Portugal; sara.baptistadasilva@gmail.com (S.B.-S.); bbernardes@ucp.pt (B.G.B.); sandraferreiraborges@gmail.com (S.B.); mpintado@porto.ucp.pt (M.P.); 2I+D Farma Group (GI-1645), Department of Pharmacology, Pharmacy and Pharmaceutical Technology, iMATUS and Health Research Institute of Santiago de Compostela (IDIS), Universidade de Santiago de Compostela, E-15782 Santiago de Compostela, Spain; 3Departamento de Biomedicina, Unidade de Bioquímica, Faculdade de Medicina, Universidade do Porto, 4200-319 Porto, Portugal; irodrigues@med.up.pt (I.R.); susy_gg@med.up.pt (S.G.-G.); raqsoa@med.up.pt (R.S.); 4i3S, Instituto de Investigação e Inovação em Saúde, Universidade do Porto, 4200-135 Porto, Portugal; rfernand@ibmc.up.pt (R.F.); mtpinto@i3s.up.pt (M.T.P.); 5IBMC, Instituto de Biologia Molecular e Celular, Universidade do Porto, 4200-135 Porto, Portugal; 6IPATIMUP, Instituto de Patologia e Imunologia Molecular, Universidade do Porto, 4200-135 Porto, Portugal; 7Faculdade de Ciências da Nutrição e Alimentação, Universidade do Porto, 4150-180 Porto, Portugal

**Keywords:** silk, sericin, in situ forming, horseradish peroxidase, hydrogel, wound healing

## Abstract

Chronic wounds are one of the most frequent complications that are associated with diabetes mellitus. The overproduction of reactive oxygen species (ROS) is a key factor in the delayed healing of a chronic wound. In the present work, we develop a novel in situ-forming silk sericin-based hydrogel (SSH) that is produced by a simple methodology using horseradish peroxidase (HRP) crosslinking as an advanced dressing for wound healing. The antioxidant and angiogenic effects were assessed in vitro and in vivo after in situ application using an excisional wound-healing model in a genetically-induced diabetic db/db mice and though the chick embryo choriollantoic membrane (CAM) assay, respectively. Wounds in diabetic db/db mice that were treated with SSH closed with reduced granulation tissue, decreased wound edge distance, and wound thickness, when compared to Tegaderm, a dressing that is commonly used in the clinic. The hydrogel also promoted a deposition of collagen fibers with smaller diameter which may have had a boost effect in re-epithelialization. SSH treatment slightly induced two important endogenous antioxidant defenses, superoxide dismutase and catalase. A CAM assay made it possible to observe that SSH led to an increase in the number of newly formed vessels without inducing an inflammatory reaction. The present hydrogel may result in a multi-purpose technology with angiogenic, antioxidant, and anti-inflammatory properties, while advancing efficient and organized tissue regeneration.

## 1. Introduction

The World Health Organization estimates that diabetes is responsible for over 1.6 million deaths each year and will be the seventh leading cause of death by 2030 [1]. Diabetic wounds, mainly chronic and/or complex wounds, are considered one of the most common complications that are associated with diabetes mellitus (DM) and contributes to the higher morbidity in patients [2]. The long healing process of a chronic wound may result in ulceration and serious infections leading, in the worst cases, to amputation. The overproduction of reactive oxygen species (ROS) is a key-factor in the delayed healing [2]. Intrinsically, there is a progressive loss of peripheral nerve fibers that is caused by high glycemic levels and ineffective vascularization. In this context, the choice of adequate dressing is of great importance and it needs to reflect the stage wound requirements, going from films/membranes (adherent or non-adherent), hydrogels, hydrocolloids, composite dressings, foams, etc. [3,4]. Currently, there are also many skin substitutes that are available in the market for clinical use [5]. However, most of these are formed from reconstituted extracellular matrices (ECM) and must undergo numerous processing steps to remain immunogenically inert. Commercial solutions may act as protective barriers, but they do not allow for subsequent skin processes, due to low bioactivity and the failure vascular permeability election. Thus, they do not provide the ideal cellular microenvironment for wound healing.

Hydrogels are one of the most promising materials in the treatment of chronic wounds, due to important requirements and properties such as: their natural moisture while absorbing extensive exudate, adherence-free coverage of underlying sensitive tissue, biodegradable and non-abrasive in the detachment, reducing pain through cooling, and the drug support for active intervention in the wound healing process. Even though there are several hydrogel-based dressings in the market, new wound care solutions are urgently needed to deal with the growing number of serious, acute, and chronic wounds that are experienced by today’s aging society.

Recently, synthetic and semi-synthetic hydrogels have been developed presenting control over the shape, physical properties, and encapsulated cargo possibilities, offering hope that more advanced products will soon reach the clinic [6]. Hydrogel formulations enclose high quantities of water and, therefore, are specific for dry or low exudative wounds, where they stimulate the cellular re-epithelization, keeping a moist environment in the wound. In this sense, several new natural-based polymeric hydrogels have been attracting wide attention in the field of wound care due to their unique structure, adaptation to the wound, and excellent biocompatibility, biodegradability, and bioactivity [7,8]. Moreover, they can incorporate a diversity of bioactive agents with the purpose of functional and responsive dressings that are able to unblock the healing process [9]. Several injectable hydrogels with antibacterial, antioxidant, anti-infection, and regenerative properties have been developed recently using biopolymers combined with specific bioactive molecules [10,11,12,13,14,15,16]. In a tissue engineering approach, hydrogels are also used for the encapsulation of therapeutic drugs or cells to support the natural healing process and to promote tissue regeneration [17]. Relevant cells have been used, such as mesenchymal stem cells-derived exosomes [18] and spheroids [19,20], and human fibroblasts cells [18,19,20,21]. However, there are questions that remain unanswered in these findings such as the adequate fit to the wound shape to avoid a loss of time and material during a dress “reshaping” by the clinician, all of which highlight the need for further developments.

Amongst the new generation of natural-based biopolymers being proposed for wound healing and regeneration, silk proteins are particularly interesting due to their exceptional properties, such as biocompatibility, oxygen and water vapor permeability, enzymatic degradability, processing versatility, and diversity of side chain chemistries that are available for ‘decoration’ [22,23]. Industrially, silk is mostly obtained from genetically controlled domestic silkworm *Bombyx mori*, with very low batch-to-batch variability. This fiber is mainly composed of silk fibroin (SF, a fibrous protein) and silk sericin (SS, a globular protein). Fibroin has been extensively studied and used for several biomaterial applications [22,23,24]. SS, about 30% of the silk cocoon, is a family of serine-rich silk proteins that glue fibroin fibers together [25]. For many years, SS has been discarded in degummed wastewater as a by-product of textile industry waste, causing a negative environmental impact. On the other hand, it was less studied than fibroin, since some works reported immune and allergenic responses when present in the fiber in its native form [23,25]. However, recent works have demonstrated that extracted SS does not present immunogenic properties, pointing it out as a promising biopolymer for biomedical and biotechnological applications. For instance, Aramwit et al. [26] showed that SS-treated wounds exhibited faster healing and had lower levels of inflammatory mediators. Nagai et al. [27] also reported the enhancing effects of SS on corneal wound healing in fatty rat. In fact, SS is already in the market for skin-related cosmetic applications. A significant amount of work in the literature highlights SS for its biodegradability, biocompatibility, antioxidant behavior, and regenerative potential [23,25,28]. A recent study showed that SS extracted by heat, acid, alkali, or urea, was able to present antibiofilm activity in wounds [29]. Additionally, it has been used to produce covalently crosslinked hydrogels for cell proliferation and drug delivery [25,30,31,32]. Nonetheless, most of these systems require conjugation with other polymers, harsh complex chemistries, or present slow gelling kinetics that are not compatible with in situ application nor biofabrication.

In this work, a novel silk sericin hydrogel (SSH) was prepared by a simple methodology using Horseradish peroxidase (HRP) as an effective enzymatic crosslinking agent [33]. This strategy allows for the development of a state-of-the-art wound dressing technology that is able to deliver in situ, i.e., directly onto the wound site, a construct that perfectly fits the size and shape of the wound and can promote healing and regeneration. Herein, the biocompatibility and antioxidant properties of an in situ-forming hydrogel dressing were analyzed through the evaluation of the local and systemic levels of antioxidant molecules, establishing a correlation between the in vitro and in vivo assessment. Moreover, it provides complementary information on the thermal stability, wettability, and transparency of the developed formulation, as well as its antimicrobial properties.

## 2. Materials and Methods

### 2.1. Materials and Reagents

*Bombyx mori* cocoons were acquired from the Sericulture of the Portuguese Association of Parents and Friends of Mentally Disabled Citizens (APPACDM, Castelo Branco, Portugal). Cocoon breading and manipulation was carefully performed under aseptic conditions according to international standardized guidelines. Horseradish peroxidase (HRP, type VI, 260  U/mg) and all other materials and reagents were purchased from Sigma-Aldrich (St. Louis, MO, USA), unless mentioned otherwise.

### 2.2. Silk Sericin Extraction and Hydrogel Preparation

SS was extracted by immersing cleaned and cut cocoon fragments (~1 cm^2^) in deionized water in a proportion of 1:100 (*w*/*v*) at 100 °C for 60 min. The SS solution was then concentrated by controlled evaporation to reach 0.7% (*w*/*v*), which was determined by the dry weight method at 105 °C for 24 h. The SS stock solution was kept at 4 °C for further experiments. To produce the hydrogel, a previously developed and optimized protocol was used (Figure 1) [33]. SS was again concentrated by controlled evaporation to a final concentration of 14% (*w*/*v*). HRP and hydrogen peroxide (H_2_O_2_) were both prepared in a phosphate buffer solution (PBS) and mixed at final concentrations of 0.010 and 0.015% (*w*/*v*), respectively.

### 2.3. Silk Sericin Hydrogel In Situ Experimental Design in a Wound Simulator

In order to study the applicability and feasibility of the SSH, an in situ simulation of a clinical treatment was made in a silicone foot ulcer simulator. Here, it was intended to study the simple application of the hydrogel in its preparation form, fast gelation, and macroscopy assessment of the detachable properties without leaving residues. The simulation was performed immediately after enzymatic crosslinking, using a 5 mL syringe and it was applied by a clinician.

### 2.4. Differential Scanning Calorimetry Analysis

The thermal analysis of SSH was performed to confirm the crosslinking effect through the increase of thermal stability of the hydrogel after crosslinking. For this purpose, a differential scanning calorimetry (DSC; model DSC-60, Mettler Toledo, Shimadzu, OH, USA), was used. The hydrogels were prepared and kept at 4 °C for 24 h, then 5.0 mg of the hydrogel was crimped in a standard aluminum pan and heated from 30 to 350 °C at a constant rate of 10 °C/min, under constant purging of nitrogen at 20 mL/min. All the samples were run in triplicate and the data that are presented were the average of the three measurements.

### 2.5. Fourier Transformed Infrared Spectroscopy with Attenuated Total Reflectance (FTIR-ATR)

FTIR-ATR analysis of Sericin solution and the formed SSH samples was performed in a spectrum series, a Perkin Elmer spectrometer (Waltham, MA, USA) that was equipped with an attenuated total reflectance (ATR) sampling accessory (PIKE Technologies, Beaconsfield, UK), and a diamond/ZnSe crystal. For each sample, a measurement of 32 scans was collected at a resolution of 4 cm^−1^, which was acquired over a wavenumber range of 600–4000 cm^−1^. In addition, baseline point adjustment and spectra normalization were performed. Water was used as a background.

### 2.6. Scanning Electron Microscopy

The surface morphology of SSH was analysed by scanning electron microscopy (SEM). SSH was freeze dried (Armfield SB4 model, Ringwood, UK) and cut to obtain middle cross-section samples, which were then sputter-coated with a thin layer (9–12 nm) of gold/palladium (SPI Module Sputter Equipment, FEI Company, Hillsboro, OR, USA) and observed using an SEM microscope (Quanta 400 FEG ESEM/EDAX Genesis X4M; FEI Company, Hillsboro, OR, USA). The micrographs were taken at an acceleration voltage of 15 kV at different magnifications.

### 2.7. Degradation Behavior and Antioxidant Activity

In order to determine the degradation behavior of SSH, the hydrogels were tested in PBS and PBS with protease XIV from *Streptomyces griseus* at a physiological concentration of 3.2 U/mg and a temperature of 37 °C, according to previous works [33]. The wet weight of the sample was measured at 0.5, 1, 2, 4, 6, and 24 h. The degradation ratio at time point was calculated using the following Equation (1):(1)$$weight\;loss\;\left(\%\right)=\frac{w_{i}-w_{f}}{w_{t}}\times 100$$
where $w_{i}$ is the initial wet weight of the hydrogel and $w_{t}$ is the wet weight that was tested at each point.

The antioxidant potential was evaluated for SSH under physiological protease (3.5 U/mg) degradation during 24 h at 37 °C. Aliquots at different time points were taken (0, 0.5, 1, 2, 4, 6, and 24 h). Then, the oxygen radical absorbance capacity (ORAC) assay was performed according to Sousa et al. [34]. The reaction was carried out at 40 °C in 75 mM phosphate buffer (pH 7.4) and the final assay mixture (200 μL) that was composed of fluorescein (70 nM), AAPH (14 mM), and Trolox (1–8 μM) or sample (at different concentrations). The fluorescence was recorded using a microplate reader (Synergy H1, Winooski, VT, USA) at excitation and emission wavelengths of 485 and 528 nm, respectively, for 140 min at intervals of 1 min. All the reaction mixtures were prepared in triplicate and final ORAC values were expressed as µmol Trolox equivalent/mL (eq Trolox µmol/mL).

### 2.8. Antimicrobial Activity

The antimicrobial activity of the SS solution was determined using an inoculum of 0.5 McFarland (1.5 × 108 CFU/mL) of each bacterium that was tested. *Staphylococcus aureus* DSM 11729, *Staphylococcus aureus* ATCC 29213, *Pseudomonas aeruginosa*, and *Escherichia coli* ATCC 25922, as the most common pathogenic contaminants in wound infection, and herein were obtained from the collection culture of CBQF, Portuguese Catholic University. The isolates were grown aerobically on nutrient agar (Merck, Taufkirchen, Germany) at 37 °C for 24 h. Muller-Hinton broth (Biokar, Allone, France) with 0.7% (*w*/*v*) SS solution was inoculated with each bacterium. A total of three controls were simultaneously assessed, one with 0.7% (*w*/*v*) of SS solution without inoculum, other with the Muller-Hinton broth with and without the inoculum. The absorbance was measured in a microplate reader (Fluostar Optima, BMG Labtech; Ortenberg, Germany) at 620 nm during 24 h at 37 °C, and the growth inhibition was analyzed by comparison with the controls. The screening of antimicrobial activity of SSH was performed by a well diffusion assay. Plates of nutrient agar were seeded with an inoculum of 0.5 McFarland (1.5 × 108 CFU/mL) of each bacterium. Wells with a diameter of 4 mm were punctured into the plates and filled with 20 μL of hydrogel. The plates were incubated for 24 h at 37 °C. The presence or absence of translucent halo zones around the wells were considered as positive or negative for antimicrobial activity, respectively. A negative control was made with Ringer solution and the assays were performed in triplicate.

### 2.9. In Vivo Chicken Embryo Choriallantoic Membrane (CAM) Assay

The CAM model was used to evaluate the angiogenic response of the SSH and its biocompatibility in comparison with a saline solution upon wounding of the membrane.

Briefly, fertilized chick (Gallusgallus) eggs that were obtained from commercial sources (Pintobar, Braga, Portugal) were incubated at 37.8 °C and 70% humidity and referred to embryonic day (E). On E3 a square window was opened in the shell after the removal of 2–2.5 mL of albumin to allow the detachment of the developing CAM. The window was sealed with a transparent adhesive tape and the eggs were returned to the incubator. At E9, two silicon rings were placed in distinct areas of the same CAM. Using a micro-scissor, a wound was inflicted in the CAM area inside the ring (~2 mm). Following, 10 µL of saline and SSH were inoculated inside each of the rings. After 4 days post-inoculation (E13) the CAMs were fixed with 4% paraformaldehyde in ovo and removed from the embryos. Ex ovo digital images of the CAM inoculation sites were acquired under a stereoscope at 20× magnification (Olympus, SZX16 coupled with a DP71 camera, Tokyo, Japan). The number of new vessels growing towards the inoculation site, delimited by the ring mark were counted (less than 20 μm in diameter) as previously described [35,36].

All the experiments using chick embryos were carried out in accordance with the Directive 2010/63/EU of the European parliament and of the council (22 September 2010) on the protection of animals used for scientific purposes, as well as the National Regulations (Decreto-Lei n.°113/2013). Accordingly, the experiments with chick embryos do not require approval from any licensing committee.

### 2.10. Animal Model of Skin Wound Healing

An excisional wound-healing model in genetically diabetic db/db mice (Charles River Laboratories Inc., Barcelona, Spain) was performed to evaluate the effect of SSH in chronic wounds. A total of 18 animals were anesthetized with inhaled isoflurane, the dorsal surface was shaved, and 70% ethanol was applied to the surgical site before the procedure. Then, two middorsal excisional wounds were created using a biopsy punch (3 mm diameter), as we previously described [37]. The animals were randomly divided into three experimental groups (n = 6 per group) as follows: Control (wounds were left to heal without any dressing); Tegaderm group (wounds covered with Tegaderm, a dressing that is commonly used in the clinic); and the SSH group (wounds were treated with SSH and covered with Tegaderm). The reason for the use of Tegaderm, was to prevent the animals from physically removing the SSH from the wound site. SSH was applied into diabetic wounds, easily adapting to wound bed, and was re-applied once a week throughout the study. The animals were maintained under controlled conditions of temperature 23 ± 5 °C and humidity of 35 ± 5% under a 12 h dark/light cycle and allowed free access to regular chow diet and water ad libitum. The animals were sacrificed with isoflurane at day 14 post-injury in the control group and 21 post-injury for the other groups. The skin tissue of one wound was collected, fixed in 10% neutral-buffered formalin, processed by dehydration incubating through a graded series of ethanol, xylol, and embedded in paraffin. The skin tissues were sectioned in 5 μm thick slices for collagen formation by Masson’s trichrome staining using a standard procedure. The skin tissue of the second wound of each animal was collected, snap frozen, and stored at −80 °C. All the animal experiments were conducted at the animal house located at the Faculty of Medicine, University of Porto, and were carried out by trained technicians in accordance with the European Community policy for Experimental Animal Studies [European Community law dated from 24 November 1986 (86/609/CEE) with addendum from 18 June 2007 (2007/526/CE)].

### 2.11. Transmission Electron Microscopy

Formalin-fixed tissues were prepared for ultrastructure analysis. Using a new razor blade, the area of interest was cut out from the paraffin block and cut into 1–2 mm cubes on filter paper. The paper with the tissue was placed in a 60 °C oven for 5–10 min or until the paraffin had melted and been absorbed by the filter paper. The tissue was transferred into a scintillation vial that was half filled with 100% xylene. At least four changes of xylene was repeated, for 15 min each, or overnight followed by two more changes of 15 min each. An angled rotating device was used. Two changes in absolute ethanol for 15 min each were processed, followed by 10 min of 95% and 10 min of 70% ethanol. Three changes of 0.1 M sodium cacodylate buffer, were repeated for 5 min each. We proceeded with standard embedding, starting with an incubation in 1% osmium tetroxide (#19190; Electron Microscopy Sciences, Hatfield, MA, USA) diluted in 0.1 M sodium cacodylate buffer for 1 h and washed in water and stained with aqueous 1% uranyl acetated solution overnight. The samples were then dehydrated and embedded in Embed-812 resin (TAAB, Berks, England). Ultra-thin sections (50 nm thickness) were cut on RMC Ultramicrotome (PowerTome, Boeckeler Instruments, Butterfield, MN, USA) using Diatome diamond knives, mounted on copper grids (DDK, Wilmington, DE, USA), and stained with uranyl acetate substitute (#11000; Electron Microscopy Sciences, Hatfield, MA, USA) and lead citrate (#11300; Electron Microscopy Sciences, Hatfield, MA, USA) for 5 min each. Thin sections were examined under a JEOL JEM 1400 transmission electron microscope (JEOL, Tokyo, Japan) and images were digitally recorded using a CCD digital camera Orius 1100 W (Tokyo, Japan).

### 2.12. Superoxide Dismutase (SOD) Activity

The SOD activity was evaluated in skin lysates that were prepared by a homogenization of skin wounded tissue (50 mg) with 2 mL of T-PER™ Tissue Protein Extraction Reagent (Merck, Taufkirchen, Germany) with a MagNA Lyser (Roche, Amadora, Portugal) followed by protein content determination by Pierce™ BCA protein assay kit (Merck, Taufkirchen, Germany) [38]. Equal amounts of skin lysate were used to determine the superoxide dismutase activity (inhibition rate %), addressed by SOD determination kit (Merck, Taufkirchen, Germany), according to the manufacturer’s instructions. Briefly, the sample solutions were mixed with water-soluble tetrazolium salt (WST) and enzyme working solutions followed by an incubation at 37 °C for 20 min. The optical density was measured at 450 nm. The results are expressed as the % inhibition rate in comparison to the control.

### 2.13. Catalase Activity

The catalase activity was assessed in skin lysates that were prepared as previously described. Then, each sample was incubated with 1 mM hydrogen peroxide (H_2_O_2_) during 30 min at 25 °C in which the catalase activity converts H_2_O_2_ into water and oxygen. The unconverted H_2_O_2_ reacted with the oxide probe and the final product was measured at 570 nm in a microplate reader [39]. The catalase activity is expressed as nmol/min/mL. The results are expressed as the means ± SD.

### 2.14. Advanced Oxidation Protein Products (AOPP) Determination

A detection of advanced oxidation protein products (AOPP) was performed spectrophotometrically in skin homogenates using the AOPP Assay Kit (Abcam, Cambridge, UK), according to the manufacturer’s instructions. Briefly, 200 µL of each sample was incubated with 10 µL of chloramine initiator solution for 5 min. After adding stop solution, the levels of AOPP were quantified through measuring the decrease in absorbance of the reaction mixture at 340 nm, in comparison to the AOPP-human serum albumin conjugate, used as positive control. The results are expressed as the means ± SD.

### 2.15. Statistical Analysis

Statistical analysis was performed comparing the three experimental groups, with one-way ANOVA multiple comparisons and Bonferroni post hoc test, using GraphPad Prism 8.0 software. In the CAM assay, a paired *t*-test was performed to compare both experimental conditions. The data are expressed as the mean ± SD. The results were considered statistically significant whenever *p* value ≤ 0.05.

## 3. Results

### 3.1. Physicochemical Characterization

Figure 2 presents the physicochemical characterization of the SSH. ATR-FTIR analysis is useful to evaluate the SSH chemical structure.

According to Figure 2A, it was possible to verify the presence of the main characteristic bands of SS that are assigned to the presence of β-sheet structure, characterized by strong bands on the amide I (1600–1690 cm^−1^) and amide II (1480–1575 cm^−1^) regions [33]. The addition of HRP and H_2_O_2_ to SS allows a fast gelation process; the free phenolic groups in sericin were fast and efficiently cross-linked within the short timeframe [33]. The structural fingerprint on the amide I region suggests that the HRP crosslinking leads to weightless alteration toward the secondary structure. In amide I, the characteristic absorption peaks of β-sheets, random coils, and α-helices are at 1630, 1645 and 1655 cm^−1^, respectively [33]. In Figure 2A, the sericin solution and SSH presented a peak at 1650 cm^−1^. This peak is typical of the predominant presence of α-helices conformation and random coils. Here, by this structural chemistry analysis, it can be verified that the functional groups were not affected after crosslinking and that the inherent bioactivity has not been compromised as previously reported.

The thermal stability of SS before and after HRP-mediated crosslinking, was evaluated by differential scanning calorimetry (DSC) in a ramp temperature up to 320 °C (Figure 2B). Before crosslinking, an endothermic peak was observed in the thermogram at 68.3 °C, while after introducing HRP in the system, the crosslinking of this endothermic peak shifted to 80.8 °C. This delay of ca.10 °C was related to the crosslink effect, which increased the thermal stability as previously reported [40]. Moreover, after a temperature ramp up to 320°, there were no thermal changes both before and after the crosslinking, which shows that the enzymatic process did not change the protein stability at high temperatures.

The SEM images (Figure 2C) show the micrography of the middle cross-section of the SSH after freeze-drying, revealing the formation of a highly interconnected porous structure (mean pore size ~50 μm), adequate for cell proliferation as well as for vascularization. This porosity feature is crucial for cell support and living towards tissue regeneration.

Figure 2D presents macroscopic images of the hydrogel being applied in situ on a wound simulator, conducted by a specialized nurse to infer on its feasibility. The hydrogel was easily applied without leaking. The gelation time for the formulation was about 3.61 ± 0.36 min, corresponding to the expected timeframe in a clinical scenario and in accordance with our previously rheological studies [33]. The hydrogel was able to fill the entire shape of the wound and had adequate elasticity and conformability. It was observed that the hydrogel had a good degree of transparency, which can allow for wound visualization during the treatment. Afterwards, it was possible to completely remove the dressing without losing its integrity.

### 3.2. Antioxidant Activity vs. Degradation Behavior

The antioxidant activity was quantified by an oxygen radical absorbance capacity (ORAC) assay, which allows for the evaluation of the scavenging capacity due to a hydrogen-atom transfer mechanism. There are several methods to determine antioxidant profiles such as: 1,1-diphenyl-2-picrylhydrazyl (DPPH), chemiluminescence, ABTS, ORAC, ferric reducing antioxidant power (FRAP), and electron spin resonance (ESR) [41]. The ORAC method assesses by measuring the elimination capacity of peroxyl radicals that are generated by the AAPH reagent and by measuring the time lapse degradation of the fluorescein probe (i.e., the rate of decrease in the intensity of fluorescence) [41].

The antioxidant activity during in vitro degradation under physiological protease concentration showed that SSH has antioxidant activity over time (Figure 3). To the best of our knowledge, this was the first study that evaluates the antioxidant potential by ORAC procedure under physiological protease (3.5 U/mg) degradation after 24 h at 37 °C. The antioxidant activity increased during the degradation, reaching a maximum level at 24 h of 3.84 eq Trolox µmol/mL. The antioxidant activity can be a vital property in controlling the oxidation of inflammatory processes that are associated with healing and tissue regeneration events. In this cellular cascade, there is an excessive production of free radicals that may damage proteins, lipids, and ECM elements and, therefore, compromises the time and success of the healing process [42].

Our results concerning an enzymatic degradation over time showed an increase of the antioxidant activity of the SSH, which may indicate that it was possible to preserve some antioxidant properties of SS after crosslinking with HRP/H_2_O_2_, also provided by other remaining unreacted phenol groups in the network. Moreover, after 24 h the maximum activity was obtained, which may be also related to the release of the phenolic groups that bind by the crosslinking, that after release may retain the radical scavenging ability. These data cannot be directly compared with other reports in which the activity is tested directly from the material and/or extracts. On the other hand, the antioxidant mechanisms of SS proteins as well as the elimination components have yet to be clarified. Still, SS is well known for having antioxidant properties which may be related to the high content of serine and threonine that are abundantly contained in SS [43]. Takechi et al. [41] clearly demonstrated the antioxidant capabilities of SS against various radicals by measuring the radical scavenging activities by four different methods. The findings of this study suggest that SS proteins may be efficiently used as potent antioxidant agents in different biomedical applications.

### 3.3. Antimicrobial Activity

The antimicrobial activity against *S. aureus* ATCC 29213, *S. aureus* DSM 11729, *P. aeruginosa,* and *E. coli* ATCC 25922 was determined. The SS solution (0.7%) and SSH did not present antimicrobial activity against important pathogens of wound infection, such as *S. aureus*, *P. aeruginosa,* and *E. coli,* since there was no bacterial growth over 24 h for all strains (Figure 4) as well as there was no inhibition halo for hydrogels (data not shown).

The results that were obtained in antimicrobial assessment of sericin solution are contradictory to some reported studies that demonstrated SS solution showing antimicrobial activity for *S. aureus* and *E. coli* [44,45,46]. Xue et al. [47] investigated the interaction mechanism of *E. coli* cells that were treated with SS and SSH for wound healing. SS demonstrated inhibition by damaging the integrity of the bacterial cell membrane of *E. coli* at concentrations of 1 to 4% [47]. In this study, SS may damage the integrity of the bacterial cell membrane, thereby eventually inhibiting the growth and reproduction of *E. coli* compared to sterile gauze. The SSH also reportedly promoted fibroblast cell proliferation and accelerated the formation of granulation tissues and neo-vessels. However, in our study, even in a close concentration of 1% (0.7%), there was no evidence of antimicrobial activity for the strains that were tested, which is in accordance with the work of Kaur et al. [48]. This team evaluated SS solutions that were extracted from different silk origins (from *B. mori*, *P. Cynthia ricini*, silkworm *A. assamensis*, and wild silkworm *A. Mylitta*). The results showed no resistance to *E. coli* growth and in the case of *B. mori* cocoon, bacterial growth was observed compared to the control. The same study reported that the antimicrobial activity of silk cocoons is rather due to residues of the chemicals that are used to isolate/purify the cocoon elements [48]. In this sense, and to create an antimicrobial-based hydrogel for wound healing, a proven antimicrobial compound may be added to the final formulation, such as silver, iodine, and polyhexamethylene biguanide [49]. Also, other concentrations of SS could be further tested for antimicrobial activity.

### 3.4. Chick Embryo Choriollantoic Membrane (CAM) Response to Sericin Hydrogel

Using an adapted wound healing CAM assay, the reaction that was induced by the application of SSH on the CAM was evaluated by quantifying the number of vessels that were recruited to the inoculation site and by evaluating the induction of an inflammatory reaction. This model is simple, quick, readily-available, and allows the study of the angiogenic potential and toxicity of a variety of samples. The CAM assay is a valuable model as an integral part of the biocompatibility testing process, thus establishing an intermediate step between in vitro and in vivo models [50,51].

To reduce the animal-dependent variability, the experiment was set in a pairwise manner, where all the eggs received both SSH and saline solution (as control). As depicted in Figure 5A–C, after implantation on the top of the CAM, SSH led to a slight increase in the number of newly formed vessels (10.86 ± 1.70 vessels), in comparison with the control (9.79 ± 1.97 vessels) (Figure 5C). Additionally, since the CAM assays allows the visualization of a possible inflammatory reaction, we verified that both the control and SHH groups had no relevant inflammatory reaction (Figure 5A,B). This qualitative analysis was then validated by the analysis of H&E-stained CAMs sections. As represented in Figure 5D,E, the histologic analysis of the tissue sections demonstrates that the application of the hydrogel did not induce an inflammatory reaction, neither did the control group.

### 3.5. Histopathological Analysis of Chronic Wound Using an Animal Model of T2DM

Based on the biocompatible behavior of our SSH formulation (demonstrated by our previously published results [33] and by the CAM assay that is presented here), and its antioxidant properties, an in vivo study using an animal model of chronic wounds was performed. This study was aimed at evaluating the skin histopathology and the antioxidant potential in the living tissue.

For the histopathological analysis of the control, Tegaderm, and SSH skin tissue, a monitorization of wound closure and Masson’s Trichrome staining were performed (Figure 6). Image analysis of the wound morphology demonstrate a faster wound closure in the control group (Figure 6A,D). As depicted in Figure 6E,F, the Tegaderm and SSH groups healed at a similar wound closure rate, but slower than non-covered wounds (control). After examining the wound histology, the re-epithelization was observed in all the experimental groups (Figure 6E,F), however, the scar tissue was more pronounced in the control group. Regarding covered wounds, the local application of SSH led to a decrease in the wound edge distance and in epidermal thickness, demonstrating that this treatment was more effective than Tegaderm alone (Figure 6H,I). In addition, normal hair follicle growth was observed in the control and SSH, when compared to Tegaderm that presents low hair follicles in the dermis. Then, the skin collagen deposition was examined by Masson’s trichrome staining in the different group samples. It was also seen that the collagen deposition was lower in the SSH group when compared to the other groups. In fact, the control and Tegaderm seems to have more collagen packaging (Figure 6G,I).

The histopathological results demonstrated that the non-treated wounds (control group) closed faster (14 days post-injury). Although wound contraction is an important step to heal a chronic wound, the wounded tissue from the control group presented scars, most probably due to the fast-wound contraction that was verified in these animals. It has been shown that persistent inflammation leads to the overproduction of various pro-inflammatory cytokines and growth factors (e.g., PDGF, TGF-β1, activin), stimulating an uncontrolled proliferation of cells in a variety of fibrotic diseases, including hypertrophic scars [52,53]. Based on our previous results [33], covered wounds, and especially those ones that were treated with SSH present lower levels of pro-inflammatory mediators at the wound bed, that may be key to minimize scar formation. Wounds that were treated with SSH closed at a similar rate when compared to the Tegaderm group, however, with reduced granulation tissue and decreased wound edge distance and wound thickness, demonstrating that this treatment seems to be more effective in relation to Tegaderm alone.

The skin samples of wounded tissue were then analyzed by TEM (Figure 7). Skin tissue from animals that were treated with Tegaderm or sericin hydrogel exhibited dermis damage when compared to the control with normal dermis distribution (Figure 7A).

In Figure 7B, it was possible to observe a huge pavement in the Tegaderm and SSH samples, in comparison with the control group. Regarding the bundles of collagen fibrils, they were observed in both longitudinal and transverse orientation in the control group (Figure 7C). In the Tegaderm samples, the collagen fibrils were observed readily oriented and randomly aligned. On the other hand, SSH collagen fibrils had different orientation but with some alignment (Figure 7C).

As depicted in Figure 7D, a uniform distribution of intact collagen fibrils and small collagen fibril bundles was noticed in the control samples. A distinct profile of results occurred in the Tegaderm and SSH groups, presenting different sizes in the diameter of the collagen fibers. The presence of lipid deposits (blue arrowheads) in Tegaderm and SSH could influence the collagen fibrils aggregation and the delay in dermis contraction.

During the healing process, there was a progressive increase in collagen fiber diameter. Initially, the collagen fibers with less diameter were drifting off in the dermis during the proliferative phase, which is essential for low wound strength. This confers an advantage for skin revascularization [52,54,55]. Remodeling the skin tissue phase appears later and involves an increase in the diameter of the collagen fiber, with more tensile strength and toughness [55]. So, based on Figure 7, the SSH-treated group displayed a closed wound with lower dermis wound strength when compared to the control group, and the collagen fibers seemed to be in the phase of alignment and orientation. In fact, reduced skin fibrosis is very important for wound healing, slowing the transition between wound healing phases, ameliorating skin remodeling and regeneration, and therefore, improving the skin quality [52,54,55]. Wounds that were treated with SSH showed a pavement of collagen fibrils with less diameter and with some spaces between the bundles of collagen (Figure 7). This decrease in collagen fibers could be explained by lower levels of IL-6 in wounded skin, which in turn, are related with lower levels of TGF-β and reduced skin fibrosis [52]. In line with this, our previous findings demonstrated that SSH was able to reduce the pro-inflammatory environment at the wound bed by a decrease in the local levels of IL-6 [33]. SSH wound dressing application could promote a more controlled inflammatory response and deposition of collagen fibers with less diameter that could be an advantage for stimulating re-epithelialization, when compared to Tegaderm alone.

### 3.6. In Vivo Assessment of Sericin-Based Hydrogel Antioxidant Potential

Growing evidence suggests that ROS and oxidative stress are important regulators of several stages of the healing process when produced in balanced amounts. However, if there is an excessive production of ROS or an impairment in ROS neutralization by endogenous defenses, it causes oxidative damage, which is the main cause of chronic wounds.

In this context, a quantification of two important antioxidant enzymatic defenses, Superoxide dismutase (SOD) and catalase activity, were performed to clarify if SSH treatment could affect the endogenous protective mechanisms against oxidative stress (Figure 8). According to Figure 8A,B, no statistical significance was found on the determination of the SOD inhibition rate and catalase activity. However, these values tend to increase in the Tegaderm group, and even more in the SSH-treated wound, when compared to the control. Additionally, the levels of advanced oxidant protein products were also explored. Based on the results illustrated in Figure 8C, SSH was able to protect wounds from protein oxidative damage, in comparison to the control group (*p* = 0.04).

According to our previous findings using an animal model of chronic wounds, the in situ application of this novel SSH decreased the levels of pro-inflammatory mediators at the wound site [33]. This is crucial since the chronic wounds fail to heal due to a perpetuation of the inflammatory stage in the wound healing process. Additionally, in response to chronic hyperglycemia, there is also a supraphysiological oxidative stress, impairing wound healing. This condition arises due to an excessive production of ROS, the deleterious oxidant molecules, coupled with ineffective antioxidant endogenous defenses that may neutralize these harmful mediators. During the management of chronic wounds, a tight control of redox signals is vital to ensure the transition between inflammation to proliferation phases of wound healing and to prevent tissue damage [53,56].

During the first hours of wound healing, neutrophils and macrophages are recruited to the wound site and release cytokines, proteases, and other mediators, including the generation of high amounts of ROS as a defense against infection. Other cells that are implicated in the healing process, namely fibroblasts are also able to produce ROS, when stimulated by cytokines and could also be produced in response to toxic molecules, leading to tissue damage and cell death [56,57]. Although low levels of ROS production and oxidative stress have beneficial regulatory effects during wound healing, the long-term exposure or excessive production at the wound bed is harmful and leads to irreversible oxidative damage of cellular macromolecules. Our results showed that SSH treatment slightly induced two important endogenous antioxidant defenses, SOD and catalase, however, without statistical significance (Figure 8). Interestingly, when we explored the effect of this wound dressing on oxidative damage, it was possible to denote a significant decrease in the content of advanced oxidation protein products at the wound bed, with SSH able to protect the wound from oxidative protein damage (Figure 8). These in vivo antioxidant results were consistent with those that were obtained in vitro for wound bed simulated conditions where the hydrogel was enzymatically degraded for 24 h, and the antioxidant potential was maintained and observed by the elimination of peroxyl radicals in the ORAC method, increasing over time up to the maximum of 24 h. These data are also in accordance with previous reports that showed the antioxidant potential of SS by different procedures and extraction methodologies [41].

Altogether, the obtained results indicate that SSH is biocompatible and a promising candidate to be applied in diabetic wounds to protect from oxidative stress and promote adequate healing. This is of paramount interest to clinicians and diabetic patients due to improvements in some of the comorbidities that are associated with non-healing wounds. These results demonstrate the potential of SSH alone. However, its activity may be further improved if the hydrogel is loaded/combined with bioactive compounds to obtain more efficient wound dressings.

## 4. Conclusions

A silk sericin hydrogel (SSH) was prepared by a simple methodology using HRP as crosslinking agent, was easy to be applied without leaking, and presented a gelation time and conformability corresponding to the expected in a clinical scenario. The CAM assay indicated that this hydrogel does not induce an inflammatory reaction and can potentially lead to an increase in the number of newly formed vessels, contributing to an efficient healing process. The in vivo studies on a diabetic mouse wound model showed that the local application of SSH led to a decrease in the wound edge distance and in epidermal thickness, demonstrating that this treatment was more effective as compared with the use of Tegaderm alone. In addition, normal hair follicle growth was observed. Covered wounds, and especially those ones that were treated with SSH, presented lower levels of pro-inflammatory mediators at the wound bed, that may be key to minimizing scar formation. In fact, reduced skin fibrosis is very important for wound healing, slowing the transition between wound healing phases, ameliorating skin remodeling and regeneration, and therefore, improving skin quality. It was also possible to denote a significant decrease in the content of advanced oxidation protein products at the wound bed, with SSH able to protect the wound from oxidative protein damage. This in vivo antioxidant capability was consistent with the obtained in vitro results for wound bed simulated conditions during the degradation tests. Altogether, the obtained results indicate that silk sericin hydrogel is biocompatible and a promising candidate to be applied in diabetic wounds to protect against oxidative stress and promote adequate healing.

## Figures and Tables

**Figure 1 biomolecules-12-00801-f001:**
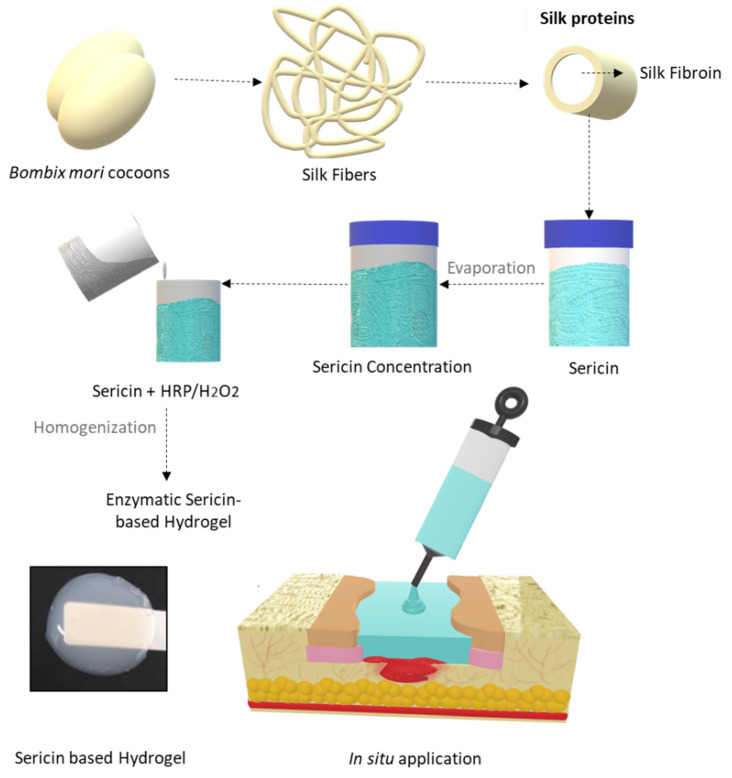
Schematic illustration of the procedure for preparing the SSH and application of the sericin-based HRP-mediated hydrogel crosslinking in a clinical setting.

**Figure 2 biomolecules-12-00801-f002:**
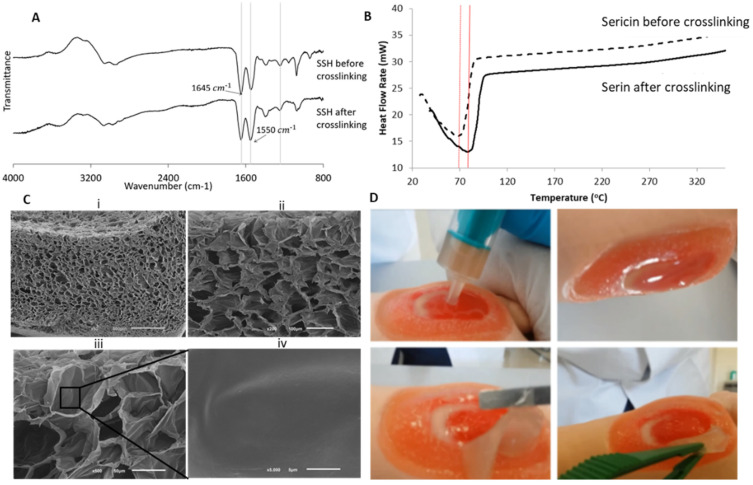
(**A**) FTIR analysis of SSH before and after crosslinking; (**B**) Thermogram of SSH before and after crosslinking; (**C**) SEM micrographs of lyophilized sericin-based hydrogels at different magnification (**i**) 50×, (**ii**) 200×; (**iii**) 500×, and (**iv**) 5000×; (**D**) SSH applied in situ on a wound simulator and macroscopic analysis.

**Figure 3 biomolecules-12-00801-f003:**
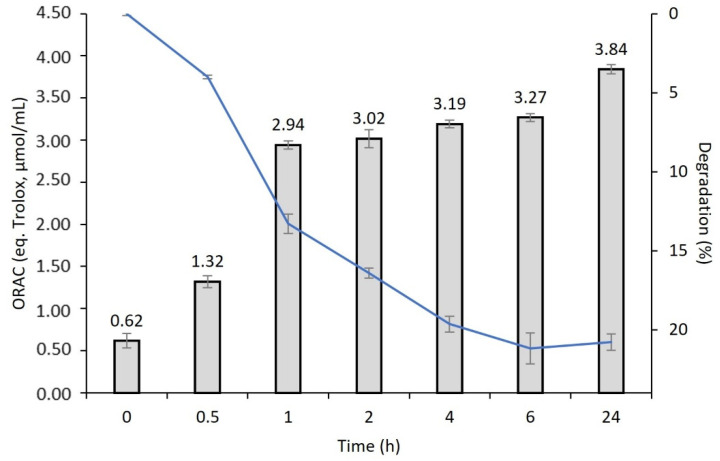
Sericin-based hydrogel antioxidant activity by the ORAC method that were obtained after physiological protease degradation during 24 h at 37 °C.

**Figure 4 biomolecules-12-00801-f004:**
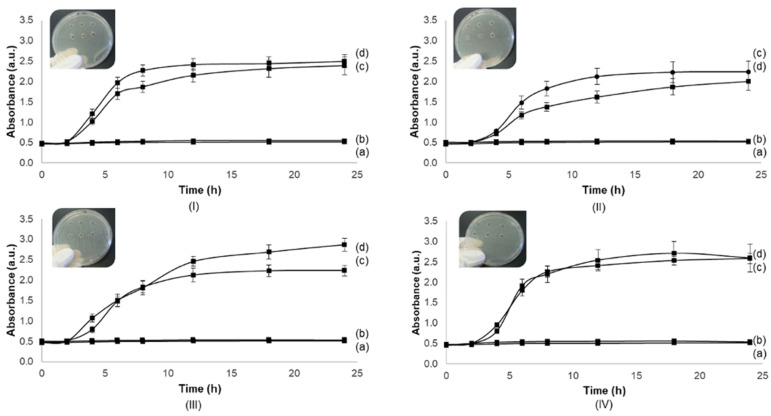
Antimicrobial activity against: (**I**) *S. aureus* ATCC 29,213, (**II**) *S. aureus* DSM 11729, (**III**) *P. aeruginosa,* and (**IV**) *E. coli* ATCC 25922 for sericin solution and for each microorganism in six replicate hydrogels (Scale bar: 9 cm) by growth curves and well diffusion method. For each figure (**I**–**IV**) letters represent: (a) negative control, sericin without inoculum; (b) negative control, Muller-Hinton broth without inoculum; (c) positive control, Muller-Hinton broth with inoculum; and (d) sericin solution (0.7%) with inoculum.

**Figure 5 biomolecules-12-00801-f005:**
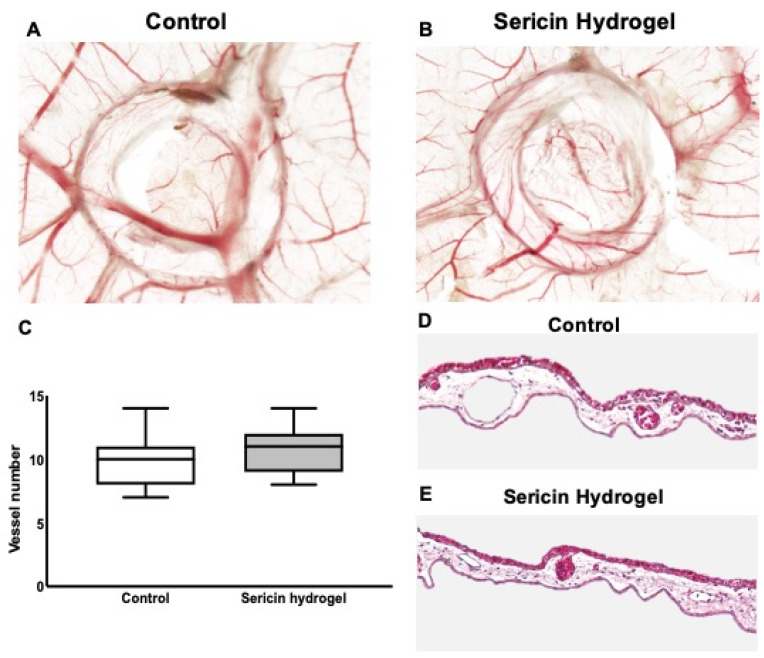
Chick embryo choriollantoic membrane (CAM) assay. (**A**,**B**) Stereomicroscope images of the excised wounded CAMs that are representative of the two conditions tested pairwise, in the same egg: Saline solution (control) and Sericin Hydrogel (SSH), four days after inoculation; (**C**) Quantification of the newly formed blood vessels (<20 μm in diameter) at the inoculation site. The data regard 14 eggs (14 pairs of samples) from 3 independent experiments (**D**,**E**) Representative images of the CAM sections that were stained with H&E, treated with Saline solution (Control), and SSH (20× magnification). The results of 15 pairs from three independent experiments are shown.

**Figure 6 biomolecules-12-00801-f006:**
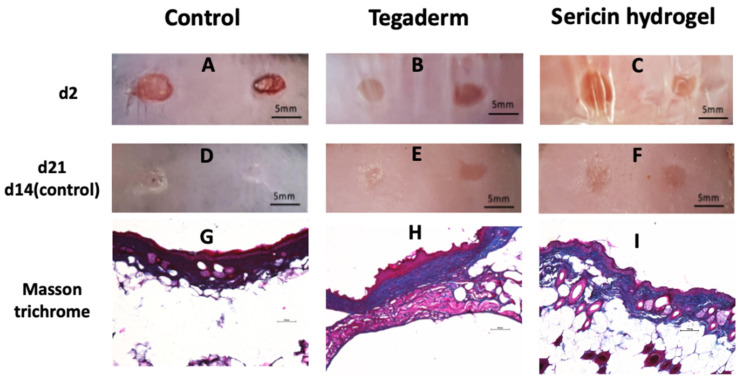
Representative images of wound closure (**A**–**F**) and histopathological analysis by Masson’s Trichrome staining (**G**–**I**) in a mouse model of a skin wound healing assay. The wounds were treated with sericin hydrogel and covered with Tegaderm, only covered with Tegaderm, or left untreated (control). Magnification of 200×. Scale bars = 200 µm.

**Figure 7 biomolecules-12-00801-f007:**
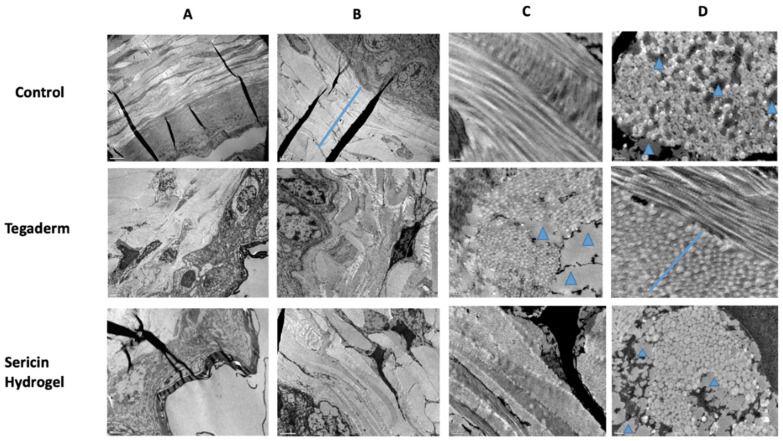
Representative images of TEM to analyze the collagen fibers at the wound bed on diabetic animals that were treated with sericin hydrogel and covered with Tegaderm, wounds only covered with Tegaderm, or wounds that were left untreated (control), in an animal model of a skin wound healing assay. (**A**) Dermis organization at low magnification (dermis-blue bar); (**B**) The network of collagen fibers; (**C**) Orientation and distribution of collagen fibrils (blue arrow head-lipid droplets; and (**D**) Diameter of collagen fibers.

**Figure 8 biomolecules-12-00801-f008:**
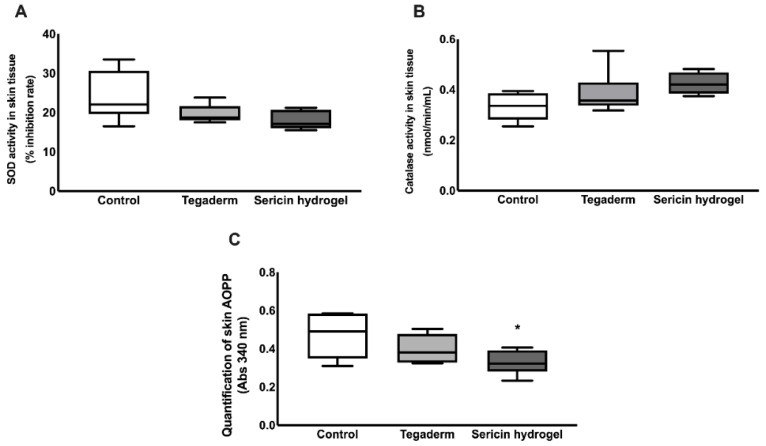
Quantification of antioxidant defenses at the wound bed on diabetic animals that were treated with sericin hydrogel and covered with Tegaderm, wounds only covered with Tegaderm, or wounds that were left untreated (control), in an animal model of a skin wound healing assay. (**A**) Superoxide dismutase (SOD) activity by enzymatic assay (expressed as % of inhibition rate); (**B**) Catalase activity, by enzymatic assay and expressed as nmol/min/mL; (**C**) Quantification of advanced oxidation protein products, assessed by a colorimetric assay. The results are expressed as the mean ± SD (n = 6) (* *p* value ≤ 0.05 in comparison to control). AOPP, Advanced Oxidant Protein Products.

## Data Availability

Not applicable.

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
