# Peer review of "Exploring Silk Sericin for Diabetic Wounds: An In Situ-Forming Hydrogel to Protect against Oxidative Stress and Improve Tissue Healing and Regeneration"

_biomolecules, 2022, doi:10.3390/biom12060801_

Round 1
Reviewer 1 Report
Dear authors,
the manuscript “Exploring silk sericin for diabetes wounds: an in situ forming hydrogel to protect against oxidative stress and improve tissue healing and regeneration” is very interesting, but requires minor revisions.
Some comments:
- The introduction is well structured, but in some places it is lacking in the literature, so I suggest the authors to increase with other articles for example: Retinoic acid/calcite micro-carriers inserted in fibrin scaffolds modulate neuronal cell differentiation, Journal of Materials Chemistry B (doi: 10.1039/c9tb01148j); Comparative assessment of cultures from oral and urethral stem cells for urethral regeneration, Current Stem Cell Research and Therapy(10.2174/1574888X10666150902094644); Green Biocompatible Method for the Synthesis of Collagen/Chitin Composites to Study Their Composition and Assembly Influence on Fibroblasts Growth, Biomacromolecules ( 10.1021/acs.biomac.1c00463); Regenerated wool keratin-polybutylene succinate nanofibrous mats for drug delivery and cells culture, Polymer Degradation and Stability(10.1016/j.polymdegradstab.2020.109272); Nano-hybrid electrospun non-woven mats made of wool keratin and hydrotalcites as potential bio-active wound dressings Nanoscale (10.1039/C8NR10114K)
- Page 8 the authors should improve the reading of the graphs in figure 2A and 2B.
- The results are described very well, I ask the authors to explain why they decided to use the embryo chicken in this way?
Author Response
Dear authors,
the manuscript “Exploring silk sericin for diabetes wounds: an in situ forming hydrogel to protect against oxidative stress and improve tissue healing and regeneration” is very interesting, but requires minor revisions.
Some comments:
- The introduction is well structured, but in some places it is lacking in the literature, so I suggest the authors to increase with other articles for example: Retinoic acid/calcite micro-carriers inserted in fibrin scaffolds modulate neuronal cell differentiation, Journal of Materials Chemistry B (doi: 10.1039/c9tb01148j); Comparative assessment of cultures from oral and urethral stem cells for urethral regeneration, Current Stem Cell Research and Therapy(10.2174/1574888X10666150902094644); Green Biocompatible Method for the Synthesis of Collagen/Chitin Composites to Study Their Composition and Assembly Influence on Fibroblasts Growth, Biomacromolecules ( 10.1021/acs.biomac.1c00463); Regenerated wool keratin-polybutylene succinate nanofibrous mats for drug delivery and cells culture, Polymer Degradation and Stability(10.1016/j.polymdegradstab.2020.109272); Nano-hybrid electrospun non-woven mats made of wool keratin and hydrotalcites as potential bio-active wound dressings Nanoscale (10.1039/C8NR10114K)
The authors are grateful for the Reviewer suggestion and have included the most relevant and current reference in the context of the introduction. "Demetra Giuri et al 2019".
- Page 8 the authors should improve the reading of the graphs in figure 2A and 2B.
The authors improved the reading quality of the graphs in Figure 2A and 2B.
- The results are described very well, I ask the authors to explain why they decided to use the embryo chicken in this way?
The authors are grateful for the appreciation and clarify that the Embryo test was used to assess the angiogenic profile of the hydrogel, as the in vivo tests were more focused on tissue regeneration. Although the chicken chorioallantoic membrane (CAM) assay is a well-established model and display some advantages, when compared to animal studies, we used this model as a previous approach to evaluate the biocompatibility and angiogenic potential of the sericin hydrogel, in comparison with a saline solution. “CAM assay is a valuable model as an integral part of biocompatibility testing process, thus establishing an intermediate step between in vitro and in vivo models.51,52”
Then, we performed a skin wound healing assay using diabetic db/db mice to mimic the diabetic wounds, in order to explore the antioxidant and regenerative effect of SSH in skin tissue.
Reviewer 2 Report
Baptista Silva et al. have synthesized, characterized and evaluated the in-situ formed silk sericin hydrogel for protection against oxidative stress and improvement of tissue healing and regeneration. This is an interesting piece of work that is planned and executed well. However, the following points need to be addressed before this paper could be accepted for publication in Biomolecules
1. Indicate at its first instance what is Tegadrem? Like "...when compared to a commercial/standard drug Tegadrem." in the abstract.
2. Optimized quantitative data for various antioxidant, angiogenic, antimicrobial, in vitro and in viva diabetic wound healing property of silk sericin-based hydrogel reported in this study should be provided in the abstract.
3. As this is a biomolecule journal, some details of composition of Bombyx mori cocoons and importance of silk sericin as well as chemical structure and basic physicochemical properties should be provided.
4. A schematic showing the work flow in terms of various segments of methodology like synthesis, characterization, and other activity studies along with especially a sub-tree in the schematic describing the animal groupings (in section 2.10) should enhance the quick grasping at glance. 5. Section 2- the details of city, state and country name for USA or city and country name for other countries should be provided for chemicals/reagents/instruments/software besides make and model for instruments and version for software used in this study.
6. Figure 1-provide some description for the last two pictures.
7. The methods described in sections 2.11, 2.12, 2.13 and 2.14 should be provided with a reference citation.
8. Figure 2-the axis labels and legends are not clear enough to aid readability.
9. Figure 4-the inset pictures within each plot are not clear. They should be enlarged and placed separately as multiple parts of the same figure.
10. Throughout the manuscript, seconds, minutes, hour, day, week, months should be abbreviated as s, min, h, d, wk, mo respectively.
11. Figure 6-the A-F should be enlarged to make the scale bar more visible for easy readability.
12. At the end of discussion, a small paragraph or a summation figure illustrating how the presence/increase of silk sericin hydrogel affected various biological properties studied in this work. This would enable a reader for quick take home points from this article.
Author Response
Baptista Silva et al. have synthesized, characterized and evaluated the in-situ formed silk sericin hydrogel for protection against oxidative stress and improvement of tissue healing and regeneration. This is an interesting piece of work that is planned and executed well. However, the following points need to be addressed before this paper could be accepted for publication in Biomolecules
- Indicate at its first instance what is Tegadrem? Like "...when compared to a commercial/standard drug Tegadrem." in the abstract.
The authors are grateful for the Reviewer suggestion and have included a specification of tegaderm in the abstract.
- Optimized quantitative data for various antioxidant, angiogenic, antimicrobial, in vitro and in viva diabetic wound healing property of silk sericin-based hydrogel reported in this study should be provided in the abstract.
The authors appreciate the suggestion and already updated the abstract. The authors did not refer the antimicrobial activity since it was not a relevant property.
- As this is a biomolecule journal, some details of composition of Bombyx mori cocoons and importance of silk sericin as well as chemical structure and basic physicochemical properties should be provided.
The authors would like to clarify that this crucial information was already available on the first work team publication: Baptista-Silva, S.; Borges, S.; Costa-Pinto, A.R.; Costa, R.; Amorim, M.; Dias, J.; Ramos, O.; Alves, P.; Granja, P.L.; Soares, R.; et al. In Situ Forming Silk Sericin-Based Hydrogel: A Novel Wound Healing Biomaterial. ACS Biomaterials Science & Engineering 2021, doi:10.1021/acsbiomaterials.0c01745.
- A schematic showing the work flow in terms of various segments of methodology like synthesis, characterization, and other activity studies along with especially a sub-tree in the schematic describing the animal groupings (in section 2.10) should enhance the quick grasping at glance.
The authors are grateful for the input and according to the suggestion, the authors included one graphical abstract to manuscript (Attached).
- Section 2- the details of city, state and country name for USA or city and country name for other countries should be provided for chemicals/reagents/instruments/software besides make and model for instruments and version for software used in this study.
The authors are grateful for the Reviewer suggestion and have included more details to equipment and reagents.
- Figure 1-provide some description for the last two pictures.
The authors already include the description of the last two pictures in figure 1.
- The methods described in sections 2.11, 2.12, 2.13 and 2.14 should be provided with a reference citation.
The authors already include some of the supported references.
- Figure 2-the axis labels and legends are not clear enough to aid readability.
The authors appreciate the alert and already changed the labels and legends.
- Figure 4-the inset pictures within each plot are not clear. They should be enlarged and placed separately as multiple parts of the same figure.
The authors would like to thank you suggestion and already enlarge the image of the figure 4.
- Throughout the manuscript, seconds, minutes, hour, day, week, months should be abbreviated as s, min, h, d, wk, mo respectively.
The authors appreciate the suggestion and already changed the figures and legends.
- Figure 6-the A-F should be enlarged to make the scale bar more visible for easy readability.
The authors appreciate the suggestion and already changed the figures and legends.
- At the end of discussion, a small paragraph or a summation figure illustrating how the presence/increase of silk sericin hydrogel affected various biological properties studied in this work. This would enable a reader for quick take home points from this article.
The authors are grateful for this input and according to this suggestion and the previous one (number 4) the authors included one graphical abstract to manuscript.

Reviewer 3 Report
This manuscript presented an in-situ forming silk sericin-based hydrogel and studied its application for wound dressing and healing. The antioxidative, antimicrobial properties of the material were analyzed. Finally, the wound healing abilities of the hydrogel was also studied. Suggested for publication after addressing the following questions.
Main:
1. Has the author tested mechanical properties of the hydrogel? As wound dressing material, sufficient stretchability is needed.
2. What is the adhesion strength between the hydrogel and skin?
3. For applying the hydrogel as wound dressing material, was there control of humidity to prevent drying?
Minor:
1. In Figure 2A, font size it too small to read.
2. In Figure 3, it is suggested to make the right axis and the line plot a different color so that it is easier to distinguish the bar graph and line plot.
Author Response
This manuscript presented an in-situ forming silk sericin-based hydrogel and studied its application for wound dressing and healing. The antioxidative, antimicrobial properties of the material were analyzed. Finally, the wound healing abilities of the hydrogel was also studied. Suggested for publication after addressing the following questions.
Main:
- Has the author tested mechanical properties of the hydrogel? As wound dressing material, sufficient stretchability is needed.
- What is the adhesion strength between the hydrogel and skin?
- For applying the hydrogel as wound dressing material, was there control of humidity to prevent drying?
The authors would like to clarify that this crucial information was already available on the first work team publication: Baptista-Silva, S.; Borges, S.; Costa-Pinto, A.R.; Costa, R.; Amorim, M.; Dias, J.; Ramos, O.; Alves, P.; Granja, P.L.; Soares, R.; et al. In Situ Forming Silk Sericin-Based Hydrogel: A Novel Wound Healing Biomaterial. ACS Biomaterials Science & Engineering 2021, doi:10.1021/acsbiomaterials.0c01745.
Minor:
- In Figure 2A, font size it too small to read.
The authors improved the reading quality of the graphs in Figure 2A.
- In Figure 3, it is suggested to make the right axis and the line plot a different color so that it is easier to distinguish the bar graph and line plot.
The authors are grateful for this input and according to this suggestion the authors improved the figure.